# Protective Effect of Phloroglucinol on Oxidative Stress-Induced DNA Damage and Apoptosis through Activation of the Nrf2/HO-1 Signaling Pathway in HaCaT Human Keratinocytes

**DOI:** 10.3390/md17040225

**Published:** 2019-04-13

**Authors:** Cheol Park, Hee-Jae Cha, Su Hyun Hong, Gi-Young Kim, Suhkmann Kim, Heui-Soo Kim, Byung Woo Kim, You-Jin Jeon, Yung Hyun Choi

**Affiliations:** 1Department of Molecular Biology, College of Natural Sciences, Dong-eui University, Busan 47340, Korea; parkch@deu.ac.kr; 2Department of Parasitology and Genetics, College of Medicine, Kosin University, Busan 49267, Korea; hcha@kosin.ac.kr; 3Department of Biochemistry, Dong-eui University College of Korean Medicine, Busan 47227, Korea; hongsh@deu.ac.kr; 4Anti-Aging Research Center, Dong-eui University, Busan 47227, Korea; 5Department of Marine Life Sciences, School of Marine Biomedical Sciences, Jeju National University, Jeju 63243, Korea; immunkim@jejunu.ac.kr (G.-Y.K.); youjinj@jejunu.ac.kr (Y.-J.J.); 6Department of Chemistry, College of Natural Sciences, Pusan National University, Busan 46241, Korea; suhkmann@gmail.com; 7Department of Biological Sciences, College of Natural Sciences, Pusan National University, Busan 46241, Korea; khs307@pusan.ac.kr; 8Biopharmaceutical Engineering Major, Division of Applied Bioengineering, College of Engineering, Dong-eui University, Busan 47340, Korea; bwkim@deu.ac.kr

**Keywords:** phloroglucinol, keratinocytes, oxidative stress, ROS, Nrf2/HO-1

## Abstract

Phloroglucinol (PG) is a component of phlorotannins, which are abundant in marine brown alga species. Recent studies have shown that PG is beneficial in protecting cells from oxidative stress. In this study, we evaluated the protective efficacy of PG in HaCaT human skin keratinocytes stimulated with oxidative stress (hydrogen peroxide, H_2_O_2_). The results showed that PG significantly inhibited the H_2_O_2_-induced growth inhibition in HaCaT cells, which was associated with increased expression of heme oxygenase-1 (HO-1) by the activation of nuclear factor erythroid 2-related factor-2 (Nrf2). PG remarkably reversed H_2_O_2_-induced excessive ROS production, DNA damage, and apoptosis. Additionally, H_2_O_2_-induced mitochondrial dysfunction was related to a decrease in ATP levels, and in the presence of PG, these changes were significantly impaired. Furthermore, the increases of cytosolic release of cytochrome *c* and ratio of Bax to Bcl-2, and the activation of caspase-9 and caspase-3 by the H_2_O_2_ were markedly abolished under the condition of PG pretreatment. However, the inhibition of HO-1 function using zinc protoporphyrin, a HO-1 inhibitor, markedly attenuated these protective effects of PG against H_2_O_2_. Overall, our results suggest that PG is able to protect HaCaT keratinocytes against oxidative stress-induced DNA damage and apoptosis through activating the Nrf2/HO-1 signaling pathway.

## 1. Introduction

Reactive oxygen species (ROS) act as important regulatory molecules of the intracellular signaling system for physiological responses. However, excessive production of ROS beyond the antioxidant capacity of cells can cause irreversible severe oxidative damage to the cells [1,2]. Such damage has been reported to correlate with the incidence of various diseases, such as inflammation, aging, and even cancer [3,4]. Despite the fact that skin has efficient mechanisms to defend against oxidative stress, accumulation of ROS overwhelming the protection from ultraviolet B (UVB) exposure has been shown to have a decisive influence on skin cell damage, including keratinocytes. The skin cells exposed to this injury should induce apoptosis in order to remove damaged cells, and there is much evidence that various enzymatic and non-enzymatic antioxidants reduce ROS-mediated apoptosis [5,6,7]. Among the intracellular organelles, mitochondria are the main source of ROS production, and are also the most vulnerable targets of ROS. Therefore, inaccurate accumulation of ROS by oxidative stress has been recognized as one of the mechanisms that lead to apoptosis following DNA damage associated with mitochondrial dysfunction [8,9]. These observations suggest that antioxidant supplementation can maintain normal skin cell function by preventing free radical accumulation through the activation of signaling pathways that block ROS production.

There is increasing evidence that various phenolic compounds derived from marine organisms can protect cells from oxidative stress by eliminating free radicals [10,11]. Among them, phloroglucinol (1,3,5-trihydroxybenzene, PG) is one of the components of phlorotannin, and is found in abundance as a polyphenol compound in *Ecklonia cava*, an edible brown alga belonging to the Laminariaceae family [12,13]. PG has been shown to have a variety of pharmacological properties, based on its potent antioxidant activity. For example, PG has been reported to inhibit the apoptosis of blood lymphocytes and splenocytes, and of lung fibroblasts induced by ionizing radiation, by inhibiting the production of intracellular ROS [14,15]. It has also been suggested that PG blocks DNA damage and the apoptosis of lung fibroblasts, keratinocytes and mouse skin by UVB, restoring the activity of antioxidant enzymes, which are associated with the elimination of ROS production [16,17]. In addition, Park and Han [18] found that PG increased the viability and insulin secretion of pancreatic β-cells, by inhibiting the oxidative stress caused by high glucose. In addition to these observations, several previous reports have been published on the in vivo and in vitro protective effects of PG on oxidative stress induced by hydrogen peroxide (H_2_O_2_), which has proven the strong antioxidative effects of PG [19,20,21]. Moreover, the antioxidant effect of PG in Parkinson’s disease model was associated with the increased expression of nuclear factor erythroid 2-related factor (Nrf2), and the activation of Nrf2-mediated antioxidant systems [22]; similarly, Nrf2 was found to be involved in the inhibition of osteoclastogenesis by this substance [23].

Nrf2 is a leucine zipper redox-sensitive transcriptional regulator that regulates the expression of antioxidant and detoxification genes, after translocation from the cytoplasm into the nucleus in response to oxidative stress [24,25]. Among the Nrf2-dependent cytoprotective enzymes, heme oxygenase-1 (HO-1) plays a critical role in heme catabolism, and produces biliverdin, ferrous iron, and carbon monoxide. This rate-limiting enzyme is activated in response to various oxidative signals, and provides adaptive and beneficial cellular responses to oxidative damage, not limited to the degradation of toxic heme released by hemoproteins [26,27]. Although there is diverse evidence that Nrf2/HO-1 signaling protects against cell death by preventing excessive ROS generation under various oxidative stress conditions, studies to date on the direct evidence that its signaling is involved in overcoming oxidative stress by PG have not been well reported. Therefore, in the present study, we investigated whether PG ameliorates oxidative stress (H_2_O_2_)-induced DNA damage and apoptosis, and whether the Nrf2/HO-1 signaling pathway is involved in this process in HaCaT human skin keratinocytes.

## 2. Results

### 2.1. PG Inhibits H_2_O_2_-Induced Cytotoxicity in HaCaT Keratinocytes

HaCaT cells were treated with various concentrations of H_2_O_2_ for 24 h, and the 3-(4,5-dimethylthiazol-2-yl)-2,5-diphenyltetrazolium bromide (MTT) assay was performed to establish the experimental conditions. Figure 1A shows that HaCaT cells treated with H_2_O_2_ showed significant decrease in cell viability in a concentration-dependent manner. Therefore, the H_2_O_2_ concentration for inducing oxidative stress was selected to be 1 mM, which showed a survival rate of about 60% or less, compared to untreated control cells. To evaluate the protective effect of PG on H_2_O_2_-induced cytotoxicity, HaCaT cells were pretreated with 25 and 50 µM PG at concentrations that did not show cytotoxicity for 1 h before treatment with 1 mM H_2_O_2_, and cultured for 24 h. Figure 1B shows that the pretreatment with PG significantly restored cell viability in a concentration-dependent manner, compared to H_2_O_2_ alone.

### 2.2. PG Activates the Nrf2/HO-1 Signaling Pathway in HaCaT Keratinocytes

To investigate whether the anti-cytotoxic effect of PG correlates with the activation of Nrf2/HO-1 signaling, the effect of PG on the expression of Nrf2 and its downstream gene HO-1 was determined. Immunoblotting results showed that the expression of HO-1 protein was slightly increased in HaCaT cells treated with H_2_O_2_, as well as PG alone (Figure 1C,D). However, the expression of HO-1 in H_2_O_2_-treated cells was significantly increased by pretreatment with PG. In addition, the enhanced expression of HO-1 by PG was associated with an increase in total protein expression of Nrf2 and its phosphorylation (p-Nrf2) at serine 40, the active form of Nrf2, whereas, the expression of Kelch-like ECH-associated protein 1 (Keap1) was reduced in a concentration-dependent manner. To test whether the protective effect of PG on H_2_O_2_-induced cytotoxicity is due to the blockade of oxidative stress associated with increased HO-1 expression, N-acetyl cysteine (NAC), a potent antioxidant, and zinc protoporphyrin IX (ZnPP), a selective inhibitor of HO-1, were pretreated. Figure 1E shows that the decrease in cell viability by H_2_O_2_ was markedly attenuated in the cells pretreated with NAC. However, when HO-1 activity was blocked using ZnPP, the inhibitory effect of PG on the cytotoxicity induced by H_2_O_2_ was significantly abolished.

### 2.3. PG Inhibits H_2_O_2_-Induced ROS Generation in HaCaT Keratinocytes

We investigated whether mitigation of the cytotoxic protective effect of PG on H_2_O_2_ in the presence of ZnPP was related to the changes in ROS production. Figure 2A,B show that the production of ROS was markedly increased in H_2_O_2_-exposed HaCaT cells; however, the accumulation of ROS in the cells pretreated with PG was significantly reduced, compared to H_2_O_2_ alone treatment. The effect of preventing ROS formation was reconfirmed using a fluorescence microscope. Consistent with the results from the flow cytometry, the increase in the DCF-DA fluorescence intensity observed in the cells treated with H_2_O_2_ was weakened by the pretreatment of PG, as shown in Figure 2C. Furthermore, in order to determine whether PG-induced HO-1 activation was involved in its inhibitory effect of ROS generation, cells were pretreated with ZnPP and PG, and then exposed to H_2_O_2_. Figure 2 shows that ZnPP abrogated the protective effect of PG on H_2_O_2_-induced ROS production; but the production of ROS was restrained under the condition that NAC was present at the same time.

### 2.4. PG Blocks H_2_O_2_-Induced DNA Damage in HaCaT Keratinocytes

We subsequently performed the following three assays to determine whether PG prevents DNA damage. The immunoblotting results in Figure 3A,B show a marked increase in γH2AX phosphorylation (p-γH2AX, at serine 139), one of the DNA strand break markers, in H_2_O_2_-stimulated cells, compared to untreated control cells; however, the increased levels of p-γH2AX by H_2_O_2_ were almost suppressed to the control level in the presence of PG. In addition, H_2_O_2_ treatment significantly increased the production of 8-hydroxy-2′-deoxyguanosine (8-OHdG) adduct, a specific marker of DNA oxidative damage, compared to the control group, but pretreatment of PG significantly reduced the production of 8-OHdG by H_2_O_2_ (Figure 3C). Furthermore, in the comet assay, another method for detecting DNA strand breaks, there was no smeared pattern of nuclear DNA in the untreated control and PG alone treated cells. However, in the H_2_O_2_-treated cells, the length of the comet tail clearly increased, which means DNA damage occurred, and in the PG pretreated cells, tail length was obviously shorter than in the H_2_O_2_-treated cells (Figure 3D). On the other hand, ZnPP abrogated some, but not all, of the protective effects of PG on the H_2_O_2_-induced DNA damages.

### 2.5. PG Suppresses H_2_O_2_-Induced Apoptosis in HaCaT Keratinocytes

4,6-diamidino-2-phenylindole (DAPI) staining, flow cytometry analysis, and agarose gel electrophoresis were performed, to investigate whether the cytoprotective effect of PG against H_2_O_2_ on HaCaT cells was related to apoptosis suppression. The fluorescent images in Figure 4A reveal that the control cells have intact nuclei, while the H_2_O_2_-treated cells show significant chromatin condensation, which is observed in the apoptosis-induced cells. However, the morphological changes are markedly attenuated in the cells pretreated with PG before the treatment with H_2_O_2_. The results of Annexin V/propidium iodide (PI) double staining also show that the pretreatment of PG significantly decreased the frequency of apoptotic cells in H_2_O_2_-stimulated cells (Figure 4B,C). In addition, in the cells treated with PG prior to H_2_O_2_ stimulation, the inhibition of colony formation by H_2_O_2_ is markedly reduced in a concentration-dependent manner (Figure 4D). Furthermore, H_2_O_2_-induced DNA fragmentation is markedly blocked in the presence of PG (Figure 4D).

### 2.6. PG Reduces H_2_O_2_-Induced Mitochondrial Dysfunction in HaCaT Keratinocytes

To examine the protective effect of PG on mitochondrial dysfunction by H_2_O_2_, mitochondrial membrane potential (MMP) and intracellular ATP levels were evaluated. According to the results of 5,5′6,6′-tetrachloro-1,1′,3,3′-tetraethyl-imidacarbocyanine iodide (JC-1) staining shown in Figure 5A,B, changes in the ratio of polarized and depolarized cell populations were observed in HaCaT cells treated with H_2_O_2_. Along with the results, the concentration of ATP in cells exposed to H_2_O_2_ was significantly decreased, compared with cells cultured in normal medium (Figure 5C). However, PG was able to prevent these changes, and the protective effects of PG were significantly abrogated in the presence of ZnPP.

### 2.7. PG Restores H_2_O_2_-Induced Alteration of the Apoptosis Regulatory Genes in HaCaT Keratinocytes

To further investigate the mechanisms of the anti-apoptotic effect of PG and the role of HO-1, we examined the effects of PG and ZnPP on the H_2_O_2_-induced changes of expression of apoptosis regulatory genes. The immunoblotting results of Figure 6A,B show that anti-apoptotic Bcl-2 protein was significantly down-regulated in H_2_O_2_-treated HaCaT cells, while the pro-apoptotic Bax protein was up-regulated. Additionally, the expression of pro-caspase-9 and -3 was markedly reduced in H_2_O_2_-treated cells as compared with the control, and the expression of cleaved poly (ADP-ribose) polymerase (PARP), a representative substrate protein degraded by activated caspase-3, was also increased. Furthermore, the expression of cytochrome *c* in H_2_O_2_-stimulated cells increased in the cytoplasmic fraction, compared to the mitochondrial fraction, indicating that cytochrome *c* was released from the mitochondria into the cytoplasm (Figure 6C,D). However, these changes by H_2_O_2_ treatment were relatively conservative in the PG-pretreated cells, and the protective potentials of PG disappeared under the condition in which the activation of HO-1 was suppressed.

## 3. Discussion

In the present study, we investigated whether PG is effective in preventing oxidative stress-induced cytotoxicity through Nrf2-mediated HO-1 activation in HaCaT keratinocytes. The results of the present study demonstrate that PG prevented H_2_O_2_-induced DNA damage and apoptosis through the rescue of mitochondrial function by blocking ROS accumulation. We also found that PG promoted activation of the Nrf2/HO-1 signaling pathway, and inhibition of HO-1 activity eliminated the protective effect of PG, suggesting that the protective effects of PG in HaCaT cells were at least HO-1 dependent.

The mechanism of Nrf2 induction depends on the inducers and cell types, but Nrf2 plays a central role in protecting cells from oxidative damage by regulating the transcriptional activity of antioxidant genes, including HO-1 [24,27,28]. Under normal physiological conditions, Nrf2 binds to the repressor protein Keap1, and is thereby sequestered in the cytoplasm. However, in order to protect against oxidative stress, phosphorylation by several kinases is necessary for the nuclear translocation of Nrf2 liberated by the degradation of Keap1 through the ubiquitin proteasome pathway [28,29]. In our results, H_2_O_2_ alone partially increases the expression of HO-1, as well as the activation of Nrf2, but their expression and activity were further markedly increased by the co-treatment with PG, compared to cells treated with H_2_O_2_ alone. Similar to the results of this study, previous studies have shown that H_2_O_2_ can enhance the expression of HO-1 [30,31], and up-regulation of HO-1 has been identified as a defense mechanism against H_2_O_2_-induced apoptosis in a variety of cell types [24,25]. Therefore, we hypothesized that the induction of HO-1 by PG could block H_2_O_2_-induced DNA damage and apoptosis, through blocking ROS generation. In this study, we investigated the effect of ZnPP on the inhibitory effect of ROS production by PG in H_2_O_2_-treated cells, and observed that ZnPP treatment weakened the ROS scavenging activity by PG. These results suggest that the protective effect of PG on H_2_O_2_-induced oxidative stress in HaCaT keratinocytes is mediated through the activation of Nrf2/HO-1 signaling. Furthermore, the beneficial effect of PG on H_2_O_2_-induced DNA damage was significantly reduced by ZnPP, which also implied that the protective role of PG on H_2_O_2_-induced cytotoxicity was dependent on HO-1. These results are in good agreement with previous results that HO-1 confers resistance to DNA damage and apoptosis induced by oxidative stress [26,27,30], and inhibition of HO-1 activity increases oxidative stress-induced cytotoxicity, and decreases the efficacy of antioxidants [31,32].

Much experimental evidence supports that excessive ROS accumulation beyond the antioxidant function of cells is one of the mechanisms leading to apoptosis associated with mitochondrial injury [1,2]. In the induction of ROS-mediated apoptosis, ROS overload causes free radical attack of the membrane phospholipid, which in turn leads to mitochondrial membrane depolarization, resulting in the loss of MMP. Subsequently, the apoptogenic factors are released into the cytoplasm from the mitochondrial intermembrane space, and the caspase cascade is activated, which could eventually trigger apoptosis. This is considered to be the onset of the intrinsic apoptosis pathway [33,34]. At the same time, mitochondrial dysfunction promotes abnormalities in the mitochondrial respiratory chain’s electron transport pathways, ultimately interfering with intracellular ATP production [33,35]. Therefore, the intracellular ATP levels can also be used as an important index to assess the homeostasis of mitochondrial energy metabolism associated with oxidative stress [36,37]. However, HO-1 activation is recognized as a cell protection mechanism against oxidative stress-mediated mitochondrial dysfunction through inactivation of the mitochondria-mediated intrinsic apoptosis pathway [26,27]. Consistent with previous studies [38,39,40], the present study shows that when cells were exposed to H_2_O_2_, MMP levels and ATP contents were significantly reduced compared to controls, whereas PG significantly reversed the H_2_O_2_-induced loss of MMP and APT. However, in the presence of ZnPP, PG-mediated repair of mitochondrial dysfunction and the decreased production of ATP were significantly abolished. These results also match well with previous results showing that the protective effects of apoptosis against oxidative stress are related to the maintenance of ATP production by the preservation of mitochondrial function [41,42]. We therefore consider that the conservation of ATP production due to the retention of mitochondrial function is one possible mechanism by which PG can preserve the cell survival pathway from oxidative stress in HaCaT keratinocytes through HO-1 activation.

Activation of caspase-9 by cytosolic release of apoptotic factors including cytochrome *c* due to the loss of MMP is a major initial step in the initiation of a caspase-dependent intrinsic apoptosis pathway [43,44]. Activation of caspase-9 sequentially activates downstream effector caspases, including caspase-3 and -7, eventually leading to cell death. This process is accompanied by degradation of the substrate proteins of effector caspases, such as PARP, as evidence that caspase-dependent apoptosis is induced [43,45]. The activation of caspase is regulated by various proteins, including Bcl-2 family members consisting of anti-apoptotic and pro-apoptotic proteins. Among the Bcl-2 family members, anti-apoptotic proteins, such as Bcl-2, are located on the outer mitochondrial membrane to prevent the release of apoptogenic factors [45,46]. On the other hand, pro-apoptotic proteins, including Bax, antagonize anti-apoptotic proteins, or translocate to mitochondrial membranes to form membrane-integrated homo oligomers that induce mitochondrial pore formation, leading to the loss of MMP, and resulting in the cytosolic release of apoptotic factors [47,48]. Therefore, the balance of apoptotic Bax family proteins to the anti-apoptotic Bcl-2 family proteins serves as a determinant to induce or inhibit the activation of the caspase cascade for the initiation of the intrinsic apoptosis pathway. Many previous studies have shown that the induction of apoptosis by H_2_O_2_ was associated with a decrease in the Bcl-2/Bax ratio and/or activation of caspases [49,50]. However, several antioxidant natural products that can protect H_2_O_2_-mediated cytotoxicity have altered this tendency [51,52]. Furthermore, it has been reported that these phenomena are also involved in the defense mechanism against oxidative stress of PG [14,18,53]. Consistent with previous findings, our results show that the decreased expression of Bcl-2 and increased expression of Bax observed in H_2_O_2_-treated HaCaT cells reverted in the presence of PG. In addition, PG administration also blocked the H_2_O_2_-induced activation of caspase-9 and -3, and the degradation of PARP. In this respect, it is suggested that PG can rescue H_2_O_2_-induced cytotoxic injury in HaCaT cells by blocking mitochondria-mediated oxidative stress and apoptosis. However, these protective effects of PG against H_2_O_2_ were markedly hindered by the inhibition of HO-1 function using an HO-1 inhibitor, consistent with other studies [38,54,55]. Therefore, further studies on the relevance of various Nrf2-dependent enzymes besides HO-1 should be attempted, but the current results suggest that the cellular protective potential of PG against oxidative stress in HaCaT cells is at least dependent on the activation of Nrf2/HO-1 signaling.

## 4. Materials and Methods 

### 4.1. Cell Culture and PG Treatment

HaCaT keratinocyte cell line was obtained from the American Type Culture Collection (Manassas, MD, USA). The cells were maintained at 37 °C in an incubator with a humidified atmosphere of 5% CO_2_, and cultured in RPMI 1640 medium (WelGENE Inc., Daegu, Korea) containing 10% (*v/v*) heat-inactivated fetal bovine serum (WelGENE Inc.), streptomycin (100 µg/mL), and penicillin (100 Units/mL). PG was purchased from Sigma-Aldrich Chemical Co. (St. Louis, MO, USA), dissolved in dimethyl sulfoxide (DMSO; Sigma-Aldrich Chemical Co.), and diluted with cell culture medium to adjust the final treatment concentrations, prior to use in the experiments.

### 4.2. Cell Viability Assay

For the cell viability study, HaCaT cells were seeded in 96-well plates at a density of 5 × 10^3^ cells per well. After 24 h incubation, the cells were incubated with PG at 25 and 50 µM or 1 mM H_2_O_2_ for another 24 h, or pre-incubated with 25 and 50 µM PG for 1 h, before 1 mM H_2_O_2_ treatment for 24 h. The cells were also treated with 10 µM ZnPP (Sigma-Aldrich Chemical Co.), a well-established HO-1 inhibitor, or 10 mM NAC (Sigma-Aldrich Chemical Co.), a ROS scavenger, for 1 h in the presence or absence of H_2_O_2_. Subsequently, the medium was removed, and 0.5 mg/ml MTT (Sigma-Aldrich Chemical Co.) was added to each well and incubated at 37 °C for 3 h. The supernatant was then replaced with an equal volume of DMSO, to dissolve the blue formazan crystals for 10 min. Optical density was measured at a wavelength of 540 nm with a microplate reader (Dynatech Laboratories, Chantilly, VA, USA). All experiments were performed in triplicate, and the results were expressed as the percentage of MTT reduction, assuming that the absorbance of control was 100%.

### 4.3. Western Blot Analysis

After being subjected to the necessary experimental treatments, the cells were harvested, washed with phosphate buffered saline (PBS), and lysed with lysis buffer (25 mM Tris-Cl (pH 7.5), 250 mM NaCl, 5 mM ethylenediaminetetraacetic acid (EDTA), 1% Nonidet-P40, 1 mM phenymethylsulfonyl fluoride, 5 mM dithiothreitol) for 30 min to extract whole-cell proteins. In a parallel experiment, mitochondrial and cytosolic proteins were extracted using a mitochondria isolation kit purchased from Active Motif (Carlsbad, CA, USA), in accordance with the instructions of the manufacturer for detecting the cytosolic release of cytochrome *c* from mitochondria. Equal amounts of protein samples (40 µg) were subjected to SDS-polyacrylamide gel electrophoresis, and then transferred onto polyvinylidene fluoride membranes (Millipore, Bedford, MA, USA). Membranes were blocked with 5% bovine serum albumin (Sigma-Aldrich Chemical Co.) for 1 h in a mixture of Tris-Buffered Saline and Tween-20 (TBST), and probed with primary antibodies (Table 1) overnight at 4 °C. Immunoreactive bands were detected using horseradish peroxidase (HRP)-conjugated secondary antibodies (GE Healthcare Pharmacia, Uppsala, Sweden), and visualized using an ECL reagent (Amersham Biosciences, Westborough, MA, USA), according to the manufacturer’s instructions. The immunoreactive bands were detected and exposed to X-ray film. Bands of Western blotting were quantified using ImageJ (Ver. 1.46; NIH, Bethesda, MD, USA) and the ratio was determined.

### 4.4. Measurement of ROS Level

To measure the formation of intracellular ROS, cells were seeded onto 6-well plates with a density of 3 × 10^5^ cells per well for 24 h, and treated with or without PG, ZnPP, or NAC for 1 h, before adding H_2_O_2_ for a further 1 h. The cells were washed twice with PBS, suspended in PBS, and stained with 10 µM DCF–DA (Sigma-Aldrich Chemical Co.) for 20 min at 37 °C. The relative fluorescence intensity of the cell suspensions was measured by flow cytometer (Becton Dickinson, Becton Dickinson, San Jose, CA, USA). For image analysis for intracellular ROS production, the cells were seeded on a coverslip loaded 6-well plate. After 24 h of plating, cells were treated with PG, ZnPP, or NAC; and 1 h later, H_2_O_2_ was added to the plate for 1 h. After washing with PBS, 10 µM DCF–DA was added to the well, and incubated at 37 °C for an additional 20 min. The fluorescence of DCF was visualized by fluorescence microscopy (Carl Zeiss, Oberkochen, Germany).

### 4.5. Comet Assay for DNA Damage

The comet assay was performed to assess the oxidative DNA damage in individual cells. After the respective treatment, the cells were detached from the culture surface, mixed with 0.75% low-melting agarose (LMA), and then spread on a fully frosted microscopic slide that was pre-coated with 0.75% normal melting agarose. After solidification of the agarose, the slides were covered with LMA, and immersed in lysis solution (2.5 M NaCl, 100 mM Na-EDTA, 10 mM Tris, 1% Triton X-100, and 10% DMSO, pH 10) for 1 h at 4 °C, to remove proteins. The slides were then placed in a gel-electrophoresis apparatus containing electrophoresis buffer (300 mM NaOH, 10 mM Na-EDTA, pH 10) for 20 min, to allow for DNA unwinding and the expression of the alkali labile damage. Thereafter, an electrical field was applied in the same buffer for 20 min at 4°C, to draw the negatively charged DNA towards the anode. After electrophoresis, the slides were rinsed gently three times with the neutralization buffer (0.4 M Tris-HCl, pH 7.5) for 10 min at 25 °C, followed by staining with 40 µg/mL EtBr (Sigma-Aldrich Chemical Co.). The slides were observed under fluorescence microscopy.

### 4.6. Determination of 8-OHdG

The BIOXYTECH^®^ 8-OHdG-EIA™ kit (OXIS Health Products Inc., Portland, OR, USA) was used for quantitative measurement of oxidative DNA damage according to the manufacturer’s instructions. Briefly, after the necessary experimental treatment, the cellular DNA was isolated using the DNA Extraction Kit (iNtRON Biotechnology Inc., Sungnam, Korea) following the manufacturer’s protocol, and quantified. The amount of 8-OHdG, a deoxyriboside form of 8-oxoGuanine, in the DNA was determined by calculation on a standard curve measured at 450 nm absorbance, using a microplate reader, according to the manufacturer’s instructions.

### 4.7. Apoptosis Assay Using Fluorescence Microscopy

To evaluate the induction of apoptosis by observing the morphological changes of nuclei, the cells were harvested and washed with PBS, and then fixed with 4% paraformaldehyde (Sigma-Aldrich Chemical Co.) in PBS for 30 min at room temperature (RT). Next, the cells were stained with 1.0 mg/ml DAPI (Sigma-Aldrich Chemical Co.) solution, a DNA-specific fluorescent dye, for 10 min at RT in the dark, and then washed twice with PBS. The degree of nuclear condensation was evaluated in the stained cells under fluorescence microscopy.

### 4.8. Apoptosis Analysis Using Flow Cytometry

The extent of apoptosis was determined by flow cytometer using Annexin V/PI double staining. In brief, the cells were re-suspended in supplied binding buffer, and then stained with FITC-conjugated annexin V and PI (BD Pharmingen, San Diego, CA, USA) at RT for 20 min in darkness, according to the manufacturer’s protocol. The fluorescence intensities of the cells were detected by flow cytometer (Becton Dickinson), and acquisition was performed using the Cell Quest Pro software (Becton Dickinson). The annexin^−^/PI^−^ cell population was considered as normal, while the annexin V-FITC^+^/PI^−^ and annexin^+^/PI^+^ cell populations were considered as indicators of apoptotic cells.

### 4.9. Detection of DNA Fragmentation

The collected cells were dissolved in lysis buffer (10 mM Tris-HCl (pH 7.4), 150 mM NaCl, 5 mM EDTA, 0.5% Triton X-100, and 0.1 mg/mL proteinase K) for 30 min at RT. DNA from the supernatant was extracted by chloroform/phenol/isoamyl alcohol (24/25/1, *v*/*v*/*v*; Sigma-Aldrich Chemical Co.), and was precipitated by ethanol. The extracted DNA was then transferred to 1.5% agarose gel containing 0.1 µg/mL EtBr, and electrophoresis was carried out at 70 V, using Tris-acetate-EDTA (pH 8.0) running buffer. DNA aliquots were observed and photographed under UV illumination (260 nm) in a BioRad gel documentation system (Bio-Rad Laboratories, Inc., Hercules, CA, USA).

### 4.10. Analysis of MMP

To measure the loss of MMP, the cells were collected, and incubated in media containing 10 µM JC-1 (Sigma-Aldrich Chemical Co.), which is a mitochondria-specific fluorescent dye, at 37 °C for 20 min at RT in darkness. After washing twice with PBS to remove unbound dye, the green (JC-1 monomers) and red (JC-1 aggregates) fluorescence ratio that monitored the proportion of mitochondrial depolarization was immediately acquired on flow cytometer, by following the manufacturer’s instructions.

### 4.11. Detection of ATP Levels

The levels of intracellular ATP were determined using a firefly luciferase-based ATP Bioluminescence Assay Kit (Roche Applied Science, Indianapolis, IN, USA), according to the manufacturer’s instructions. Briefly, cells cultured under various conditions were lysed with the provided lysis buffer, and the collected supernatants were mixed with an equal amount of luciferase agent, which catalyzed the light production from ATP and luciferin. The emitted light was immediately measured using a microplate luminometer, and the ATP level was calculated according to the ATP standard curve. Intracellular ATP levels were calculated as a percentage of the untreated control.

### 4.12. Colorimetric Assay of Caspase-3 Activity

The activity of the caspase-3 was determined using a colorimetric assay kit (R&D Systems, Minneapolis, MN, USA), following the manufacturer’s protocol. Briefly, at the end of treatment, the collected cells were lysed, and equal amounts of proteins were incubated with the supplied reaction buffer, containing dithiothreitol and tetrapeptides (Asp-Glu-Val-Asp (DEAD)-p-nitroaniline (pNA)) as caspase-3 substrates at 37 °C in the dark. After 2 h of reaction, the reactions were measured by changing the absorbance at 405 nm using a microplate luminometer, as per the manufacturer’s protocols. The results were represented as multiples of untreated control cells.

### 4.13. Statistical Analysis

Data were expressed as the means ± SD from at least three independent experiments. Statistical significance analysis was carried out using Student’s *t*-test or ANOVA. A *p*-value less than 0.05 were considered to indicate statistical significance.

## 5. Conclusions

In summary, the present study demonstrates that PG can effectively protect HaCaT keratinocytes from H_2_O_2_-induced cytotoxicity, by blocking oxidative stress-mediated DNA damage and mitochondria-dependent apoptotic pathway through the activation of Nrf2/HO-1 signaling. Although studies of mitochondrial damage-associated energy metabolism and PG downstream signal molecules are needed, these findings may be presented as evidence that PG can alter the redox state of cells, and thereby regulate cellular antioxidant signaling pathways. Therefore, additional studies should be conducted to determine whether pathways other than the mitochondria-mediated pathway may be involved in inducing apoptosis by oxidative stress, including in vivo animal experiments.

## Figures and Tables

**Figure 1 marinedrugs-17-00225-f001:**
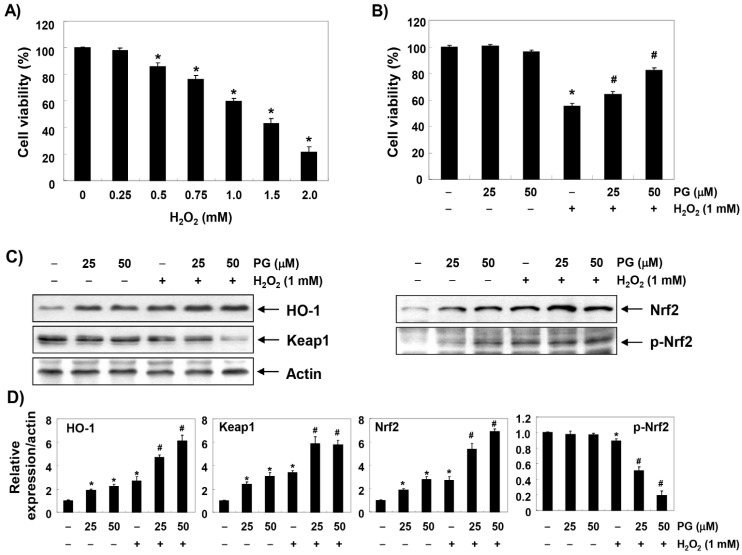
The protective effects of PG against H_2_O_2_-induced cytotoxicity in HaCaT cells. Cells were treated with various concentrations of H_2_O_2_ for 24 h (**A**) or treated with the indicated concentrations of PG for 24 h, or treated with 1 mM H_2_O_2_ for 24 h after pre-treatment with PG, 10 mM N-acetyl cysteine (NAC), and/or 10 µM zinc protoporphyrin IX (ZnPP) for 1 h (**B**–**E**). (**A**,**B**,**E**) After treatment, cell viability was examined by the 3-(4,5-dimethylthiazol-2-yl)-2,5-diphenyltetrazolium bromide (MTT) assay. Data are expressed as the mean ± standard deviation (SD) of three independent experiments. Statistical significance analysis was carried out using Student’s *t*-test or ANOVA (* *p* < 0.05 compared with the control group, ^#^
*p* < 0.05 compared with the H_2_O_2_-treated group, ^&^
*p* < 0.05 compared with the PG- and H_2_O_2_-treated group). (**C**) The cellular proteins were prepared, and the levels of HO-1, Nrf2, p-Nrf2 and Keap1 expression were assayed by Western blot analysis using an enhanced chemiluminescence (ECL) detection system. Actin was used as an internal control. These images are representative of at least three independent experiments. (**D**) Quantification of the ratios of band intensity of HO-1, Nrf2, p-Nrf2, and Keap1 relative to actin. The data were shown as mean ± SD obtained from three independent experiments. Statistical significance analysis was carried out using Student’s *t*-test or ANOVA (* *p* < 0.05 compared with the control group, ^#^
*p* <0.05 compared with the H_2_O_2_-treated group, ^&^
*p* < 0.05 compared with the PG- and H_2_O_2_-treated group).

**Figure 2 marinedrugs-17-00225-f002:**
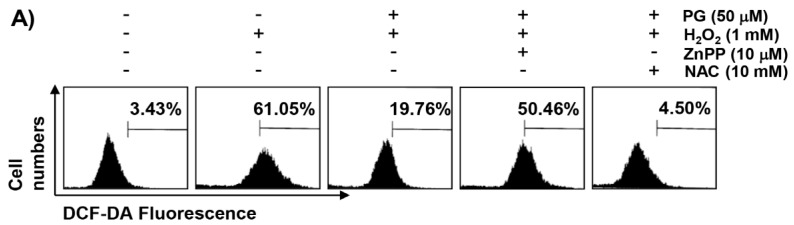
Inhibition of H_2_O_2_-induced ROS generation by PG in HaCaT cells. Cells were pretreated with 50 µM PG, 10 µM ZnPP, and/or 10 mM NAC for 1 h, and then treated with 1 mM H_2_O_2_ for 1 h. (**A**) After staining with 2,7′,7-dichlorofluorescein diacetate (DCF-DA) fluorescent dye, DCF fluorescence was monitored by a flow cytometer. (**B**) The data were shown as mean ± SD obtained from three independent experiments. Statistical significance analysis was carried out using Student’s *t*-test or ANOVA (* *p* < 0.05 compared with the control group, ^#^
*p* < 0.05 compared with the H_2_O_2_-treated group, ^&^
*p* < 0.05 compared with the PG- and H_2_O_2_-treated group). (**C**) The fluorescent images were obtained by fluorescence microscope (original magnification, ×200). These images are representative of at least three independent experiments.

**Figure 3 marinedrugs-17-00225-f003:**
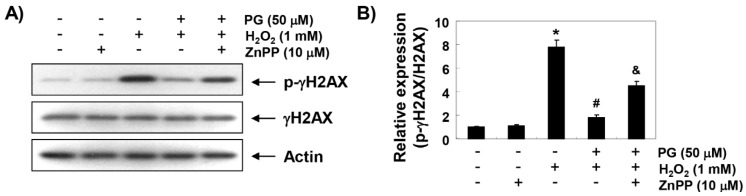
Protection of H_2_O_2_-induced DNA damage by PG in HaCaT cells. Cells were pretreated with 50 µM PG and/or 10 µM ZnPP for 1 h, and then stimulated with or without 1 mM H_2_O_2_ for 24 h. (**A**) The cellular proteins were prepared, and p-γH2AX and γH2AX protein levels were assayed by Western blot analysis, using an ECL detection system. Actin was used as an internal control. (**B**) Quantification of the ratios of band intensity of p-γH2AX relative to γH2AX. (**C**) The amount of 8-hydroxy-2′-deoxyguanosine (8-OHdG) in DNA was determined using an 8-OHdG-enzyme immunoassay (EIA) kit. The measurements were made in triplicate, and values are expressed as the mean ± SD. Statistical significance analysis was carried out using Student’s *t*-test or ANOVA (* *p* < 0.05 compared with the control group, ^#^
*p* < 0.05 compared with the H_2_O_2_-treated group, ^&^
*p* < 0.05 compared with the PG- and H_2_O_2_-treated group). (**D**) The comet assay was performed, and representative images of comet assay were taken by fluorescence microscope (original magnification, ×200). Representative photographs are shown.

**Figure 4 marinedrugs-17-00225-f004:**
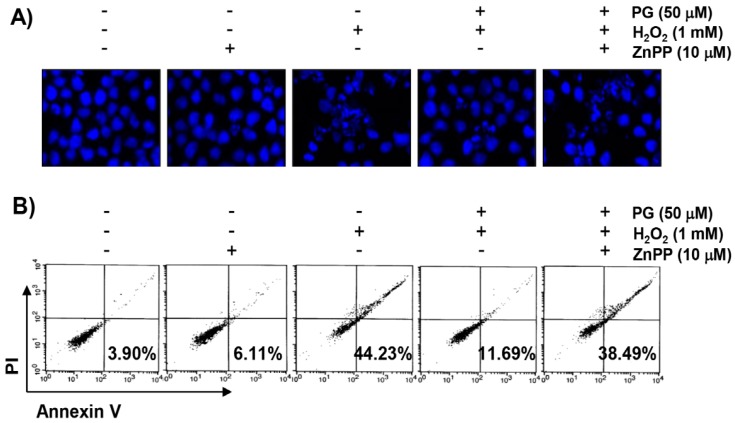
The inhibitory effects of PG against H_2_O_2_-induced apoptosis in HaCaT cells. Cells were pretreated with 50 µM PG and/or 10 µM ZnPP for 1 h, and then stimulated with or without 1 mM H_2_O_2_ for 24 h. (**A**) The cells were fixed and stained with 4,6-diamidino-2-phenylindole (DAPI) solution. The stained nuclei are pictured under fluorescence microscope (original magnification, ×400). Representative photographs are shown. (**B**) The cells were stained with fluorescein isothiocyanate (FITC)-conjugated Annexin V and propidium iodide (PI) for flow cytometry analysis. The percentages of apoptotic cells were determined by counting the percentage of Annexin V-positive cells. (**C**) Data are expressed as the mean ± SD of three independent experiments. Statistical significance analysis was carried out using Student’s *t*-test or ANOVA (* *p* < 0.05 compared with the control group, ^#^
*p* < 0.05 compared with the H_2_O_2_-treated group, ^&^
*p* < 0.05 compared with the PG- and H_2_O_2_-treated group). (**D**) DNA fragmentation was analyzed by extracting genomic DNA, electrophoresis in 1.5 % agarose gel, and then visualizing by ethidium bromide (EtBr) staining.

**Figure 5 marinedrugs-17-00225-f005:**
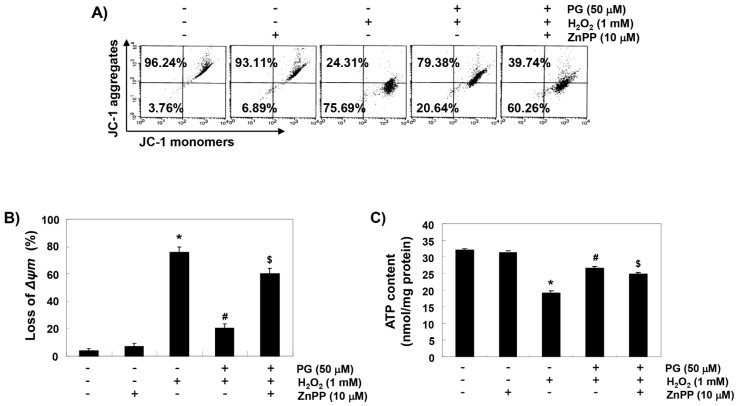
Attenuation of H_2_O_2_-induced mitochondrial dysfunction by PG in HaCaT cells. Cells were pretreated with 50 µM PG and/or 10 µM ZnPP for 1 h, and then stimulated with or without 1 mM H_2_O_2_ for 24 h. (**A**) The cells were collected and incubated with 10 µM of 5,5′6,6′-tetrachloro-1,1′,3,3′-tetraethyl-imidacarbocyanine iodide (JC-1), and the values of mitochondrial membrane potential (MMP) were evaluated by flow cytometer. (**B**) The results are the mean ± SD obtained from three independent experiments. (**C**) A commercially available kit was used to monitor the ATP production using a luminometer. Each point represents the mean ± SD of three independent experiments. Statistical significance analysis was carried out using Student’s *t*-test or ANOVA (* *p* < 0.05 compared with the control group, ^#^
*p* < 0.05 compared with the H_2_O_2_-treated group, ^&^
*p* < 0.05 compared with the PG- and H_2_O_2_-treated group).

**Figure 6 marinedrugs-17-00225-f006:**
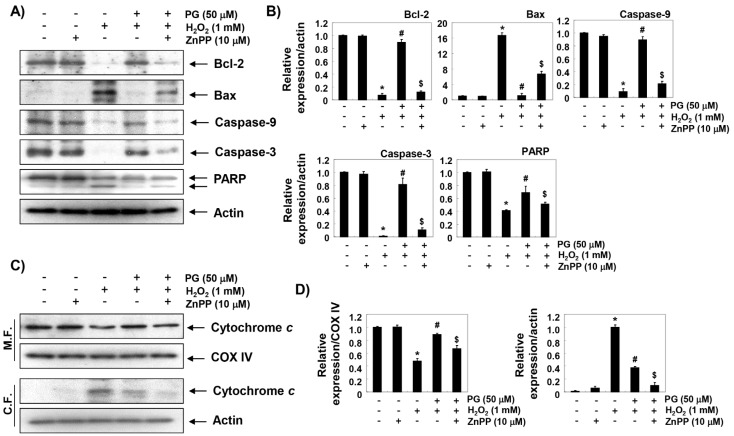
Effect of PG on the expression of apoptosis regulatory genes in H_2_O_2_-treated HaCaT cells. (**A**) After treatment with 50 µM PG and/or 10 µM ZnPP in the presence or absence of 1 mM H_2_O_2_ for 24 h, the cellular proteins were prepared, and the protein levels were assayed by Western blot analysis using an ECL detection system. Actin was used as an internal control. (**B**) Quantification of the ratios of band intensity of Bcl-2, Bax, caspase-9, caspase-3, and PARP relative to actin. (**C**) The mitochondrial and cytosolic proteins isolated from cells cultured under the same conditions were separated by sodium-dodecyl sulfate (SDS)-polyacrylamide gel electrophoresis, and transferred to the membranes. The membranes were probed with anti-cytochrome *c* antibody. Equal protein loading was confirmed by the analysis of oxidase subunit IV (COX IV) and actin in each protein extract (M.F., mitochondrial fraction; C.F., cytosolic fraction). Representative photographs are shown. (**D**) Quantification of the ratios of band intensity of cytochrome *c* relative to COX IV or and actin, respectively. Statistical significance analysis was carried out using Student’s *t*-test or ANOVA (* *p* < 0.05 compared with the control group, ^#^
*p* < 0.05 compared with the H_2_O_2_-treated group, ^&^
*p* < 0.05 compared with the PG- and H_2_O_2_-treated group).

**Table 1 marinedrugs-17-00225-t001:** Antibodies used in the present study.

Antibody	Manufacturer	Item No.	Dilution
HO-1	Merck Millipore	374090	1:1000
Nrf2	Santa Cruz Biotechnology, Inc.	sc-13032	1:1000
p-Nrf2	Abcam, Inc.	ab76026	1:1000
Keap1	Santa Cruz Biotechnology, Inc.	sc-15246	1:1000
p-γH2AX	Cell Signaling Technology, Inc.	9718	1:500
γH2AX	Cell Signaling Technology, Inc.	7631	1:500
Bcl-2	Cell Signaling Technology, Inc.	sc-509	1:1000
Bax	Cell Signaling Technology, Inc.	sc-493	1:1000
Caspase-9	Santa Cruz Biotechnology, Inc.	sc-7885	1:1000
Caspase-3	Santa Cruz Biotechnology, Inc.	sc-7272	1:1000
Cytochrome *c*	Santa Cruz Biotechnology, Inc.	sc-7159	1:500
COX IV	Santa Cruz Biotechnology, Inc.	sc-376731	1:1000
Actin	Santa Cruz Biotechnology, Inc.	sc-47778	1:1000

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
