# Peer review of "Protective Effect of Phloroglucinol on Oxidative Stress-Induced DNA Damage and Apoptosis through Activation of the Nrf2/HO-1 Signaling Pathway in HaCaT Human Keratinocytes"

_marinedrugs, 2019, doi:10.3390/md17040225_

Reviewer 1 Report

Please clarify your writing:

1. Line 94-95: When 1mM H2O2 was used, and there was a survival rate of about 60%, as you are aware that this level of oxidative stress can also induce Nrf2 and its downstream expression. Please clarify in your writing how do you take care of it an d to differentiate it from the protective effects of PG.

2. Line 96-99 - Please be consistent and clear about whether the PG was pre-treated or follow 1mM H2O2. As your 4.2 Cell Viability Assay also mentions different ways of treatment in lines 325-331. Please make clear in your results and methods to be consistent and clear in other section such as ROS measurement etc.

3. Mistakes in Line 196--Are you referring to Figure 5A etc instead of 6A and B?

4. Please spell out in full first before any abbreviation is used, eg. Line 440 MMP etc. 

Author Response

Response to Reviewer 1 Comments

Please clarify your writing:

1. Line 94-95: When 1mM H2O2 was used, and there was a survival rate of about 60%, as you are aware that this level of oxidative stress can also induce Nrf2 and its downstream expression. Please clarify in your writing how do you take care of it an d to differentiate it from the protective effects of PG.

Response 1: We are appreciated for your suggestions on our manuscript. According to your comments, MTT assay results for H2O2 alone group were added. In addition, we corrected the corresponding parts as described in the answer to the following question.

2. Line 96-99 - Please be consistent and clear about whether the PG was pre-treated or follow 1mM H2O2. As your 4.2 Cell Viability Assay also mentions different ways of treatment in lines 325-331. Please make clear in your results and methods to be consistent and clear in other section such as ROS measurement etc.

Response 2: Sorry for the confusion. These were errors that occurred during the preparation of the manuscript, and "2-1. PG Inhibits H2O2-induced Cytotoxicity in HaCaT Keratinocytes" section of the "Results". The sentences have been modified as follows.

→ HaCaT cells were treated with various concentrations of H2O2 for 24 h, and the 3-(4,5-dimethylthiazol-2-yl)-2,5-diphenyltetrazolium bromide (MTT) assay was performed to establish the experimental conditions. Figure 1A shows that HaCaT cells treated with H2O2 showed significant decrease in cell viability in a concentration-dependent manner. Therefore, the H2O2 concentration for inducing oxidative stress was selected to be 1 mM, which showed a survival rate of about 60% or less, compared to untreated control cells. To evaluate the protective effect of PG on H2O2-induced cytotoxicity, HaCaT cells were pretreated with 25 and 50 µM PG at concentrations that did not show cytotoxicity for 1 h before treatment with 1 mM H2O2, and cultured for 24 h. Figure 1B shows that the pretreatment with PG significantly restored cell viability in a concentration-dependent manner, compared to H2O2 alone.

3. Mistakes in Line 196--Are you referring to Figure 5A etc instead of 6A and B?

Response 3: These are errors that occurred during the preparation of the manuscript, which have been corrected.

4. Please spell out in full first before any abbreviation is used, eg. Line 440 MMP etc.

Response 4: According to your comments, the manuscript of the relevant parts has been revised.

Thanks to the positive evaluation and the careful comments on this paper.

Reviewer 2 Report

In this manuscript, the authors investigated the protective efficacy of phloroglucinol (PG) in HaCaT human skin keratinocytes stimulated with hydrogen peroxide to induce oxidative stress. They demonstrated that  PG is effective in preventing oxidative stress-induced cytotoxicity through Nrf2-mediated HO-1 activation in keratinocytes. The work has some weaknesses that should be addressed before publication. Here my comments:

- HaCaT cells were treated with 25 and 50 μM PG at concentrations that did not show cytotoxicity for 1 h before treatment with 1 mM H2O2, and cultured for 24 h.  On the contrary, in the figure legend 1, authors stated: Cells were 116 treated with 25 or 50 μM PG for 24 h, or treated with 1 mM H2O2 for 24 h after pre-treatment with PG.... The experimental conditions are different, please clarify.

- The up-regulation of HO-1 after treatment with PG and H2O2 seems modest. WB data should be quantified by densitometric analysis.

- In the fluorescence microscope observation, we also confirmed that PG had a strong  ROS scavenging effect (Figure 2C).  The results of this exp should be explained in the text.

Figure 2 shows that ZnPP abrogated.... Which figure panel? Clarify. 

- Antibodies code, final working dilutions, and the amount of proteins separated by SDS-Page should be reported.

- Statistical significance analysis was carried out using Student’s t-test or ANOVA.  In each figure legend, the authors should clarify which statistical test was used.

Author Response

Response to Reviewer 2 Comments

In this manuscript, the authors investigated the protective efficacy of phloroglucinol (PG) in HaCaT human skin keratinocytes stimulated with hydrogen peroxide to induce oxidative stress. They demonstrated that PG is effective in preventing oxidative stress-induced cytotoxicity through Nrf2-mediated HO-1 activation in keratinocytes. The work has some weaknesses that should be addressed before publication. Here my comments:

Response1: We are appreciated for your suggestions on our manuscript. According to your comments, the manuscript has been revised as follows.

- HaCaT cells were treated with 25 and 50 μM PG at concentrations that did not show cytotoxicity for 1 h before treatment with 1 mM H2O2, and cultured for 24 h. On the contrary, in the figure legend 1, authors stated: Cells were 116 treated with 25 or 50 μM PG for 24 h, or treated with 1 mM H2O2 for 24 h after pre-treatment with PG.... The experimental conditions are different, please clarify.

Response: Sorry for the confusion. These were errors that occurred during the preparation of the manuscript, and "2-1. PG Inhibits H2O2-induced Cytotoxicity in HaCaT Keratinocytes" section of the "Results". The sentences have been modified as follows.

→ HaCaT cells were treated with various concentrations of H2O2 for 24 h, and the 3-(4,5-dimethylthiazol-2-yl)-2,5-diphenyltetrazolium bromide (MTT) assay was performed to establish the experimental conditions. Figure 1A shows that HaCaT cells treated with H2O2 showed significant decrease in cell viability in a concentration-dependent manner. Therefore, the H2O2 concentration for inducing oxidative stress was selected to be 1 mM, which showed a survival rate of about 60% or less, compared to untreated control cells. To evaluate the protective effect of PG on H2O2-induced cytotoxicity, HaCaT cells were pretreated with 25 and 50 µM PG at concentrations that did not show cytotoxicity for 1 h before treatment with 1 mM H2O2, and cultured for 24 h. Figure 1B shows that the pretreatment with PG significantly restored cell viability in a concentration-dependent manner, compared to H2O2 alone.

- The up-regulation of HO-1 after treatment with PG and H2O2 seems modest. WB data should be quantified by densitometric analysis.

Response: According to your comments, we reviewed all of our results of Western blotting. We analyzed and quantified bands of Western blotting in previous figures and all new figures in revised manuscript.

- In the fluorescence microscope observation, we also confirmed that PG had a strong ROS scavenging effect (Figure 2C). The results of this exp should be explained in the text.

Response: Thank you for your comments. The sentence has been modified as follows.

→ The effect of preventing ROS formation was reconfirmed using a fluorescence microscope. Consistent with the results from the flow cytometry, the increase in the DCF-DA fluorescence intensity observed in the cells treated with H2O2 was weakened by the pretreatment of PG, as shown in Figure 2C.

- Figure 2 shows that ZnPP abrogated.... Which figure panel? Clarify.

Response: The part pointed out by reviewer is the content of all the results presented in Figure 2.

- Antibodies code, final working dilutions, and the amount of proteins separated by SDS-Page should be reported.

Response: According to your suggestion, the information on the antibodies used in this experiment was shown in the Table 1 and amount of proteins used in Western blot analysis was indicated in the Materials and Methods section.

- Statistical significance analysis was carried out using Student’s t-test or ANOVA. In each figure legend, the authors should clarify which statistical test was used.

Response: Thank you for commenting. The manuscript of the relevant parts has been corrected.

Again, thanks to the positive evaluation and the careful comments on this paper.

Round  2

Reviewer 2 Report

The paper has been improved and it is now suitable for publication.

Mar. Drugs EISSN 1660-3397 Published by MDPI AG, Basel, Switzerland RSS E-Mail Table of Contents Alert
Back to Top